# Perceptions of the adult US population regarding the novel coronavirus outbreak

SarahAnn M. McFadden[1,2], Amyn A. Malik[1,2], Obianuju G. Aguolu[1,2], Kathryn S. Willebrand[1,3], Saad B. Omer[1,2,3,4]*

1 Yale Institute for Global Health, New Haven, Connecticut, United States of America, 2 Department of Internal Medicine, Infectious Disease, Yale School of Medicine, New Haven, Connecticut, United States of America, 3 Yale School of Public Health, New Haven, Connecticut, United States of America, 4 Yale School of Nursing, New Haven, Connecticut, United States of America

* saad.omer@yale.edu

## Abstract

The Coronavirus Disease 2019 (COVID-19) outbreak is spreading globally. Although COVID-19 has now been declared a pandemic and risk for infection in the United States (US) is currently high, at the time of survey administration the risk of infection in the US was low. It is important to understand the public perception of risk and trust in sources of information to better inform public health messaging. In this study, we surveyed the adult US population to understand their risk perceptions about the COVID-19 outbreak. We used an online platform to survey 718 adults in the US in early February 2020 using a questionnaire that we developed. Our sample was fairly similar to the general adult US population in terms of age, gender, race, ethnicity and education. We found that 69% of the respondents wanted the scientific/public health leadership (either the CDC Director or NIH Director) to lead the US response to COVID-19 outbreak as compared to 14% who wanted the political leadership (either the president or Congress) to lead the response. Risk perception was low (median score of 5 out of 10) with the respondents trusting health professionals and health officials for information on COVID-19. The majority of respondents were in favor of strict infection prevention policies to control the outbreak. Given our results, the public health/scientific leadership should be at the forefront of the COVID-19 response to promote trust.

## Introduction

The current novel coronavirus outbreak, COVID-19, has spread across the globe with hundreds of thousands infected and thousands of deaths. [1] At the time of this writing (March 25, 2020), there are over 54,000 cases of COVID-19 in the US with 737 deaths. [2] With COVID-19 now declared a pandemic by the World Health Organization, [3] it is important to understand risk perceptions about COVID-19 and trust in political and public health/scientific leadership among the US population to better inform messaging and policies. [4]

### Objective

In the first study of its kind on COVID-19, our objective was to survey the adult US population to better understand their risk perceptions about the COVID-19 outbreak.

**Competing interests:** The authors have declared that no competing interests exist.

## Methods

Data were collected using an electronic questionnaire via Qualtrics® (Qualtrics, Provo, UT). Participants completed the questionnaire through the CloudResearch [5] online platform in early February 2020. We asked participants to rank who they felt should lead the US response to COVID-19. Options included the president, Congress, the Director of the Centers for Disease Control and Prevention (CDC), and the Director for the National Institutes of Health (NIH; S1 Survey). In addition, participants completed the perceived risk scale (Cronbach's $\alpha$ = 0.71) which had 10 survey-items (5-point Likert Scale: 0 = strongly disagree/disagree/neutral; 1 = agree/strongly agree). We also asked about their support for restrictive infection prevention policies and the reliability of various sources of information (S1 Survey). Descriptive analyses were conducted. Yale University Institutional Review Board approved this study (IRB protocol number: 2000027402). Participants provided informed consent prior to data collection.

## Results

The sample consisted of 718 adults and was similar to the US population in terms of age, gender, race, ethnicity, and education (Table 1).

**Table 1. Demographic characteristics of sample compared to US population.**

|  | Total (N = 718) n (%) | US Population* (%) |
|---|---|---|
| Gender |  |  |
| Male | 330 (46) | 49 |
| Female | 386 (54) | 51 |
| Other | 2 (0) |  |
| Age (years)** |  |  |
| 18–25 | 84 (12) | 12 |
| 26–35 | 145 (20) | 18 |
| 36–45 | 166 (23) | 16 |
| 46–55 | 111 (15) | 17 |
| 55+ | 212 (30) | 36 |
| Race |  |  |
| Black/African American | 111 (15) | 13 |
| American Indian/Alaska Native | 35 (5) | 1 |
| Asian | 69 (10) | 5 |
| Native Hawaiian/Other Pacific Islander | 2 (0) | 0 |
| White | 501 (70) | 73 |
| Ethnicity |  |  |
| Hispanic | 107 (15) | 18 |
| Non-Hispanic | 611 (85) | 82 |
| Education |  |  |
| No high school | 6 (1) | 12 |
| High school | 182 (25) | 27 |
| Some College | 174 (24) | 29 |
| College | 223 (31) | 19 |
| Graduate/Professional | 133 (19) | 12 |

*American Community Survey 2018 (5-year estimate)

** Percentages are out of total population 18 years and older

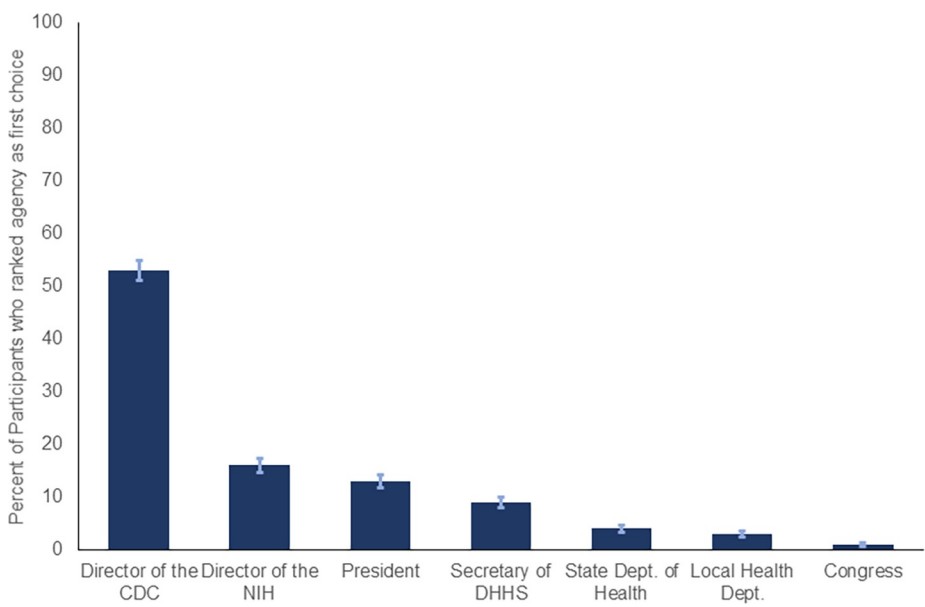

**Fig 1. Participants choice for who should lead the US response to COVID-19 outbreak.**

Over 90% of our sample was aware of the COVID-19 mostly through the news (n = 522, 73%). The majority of participants wanted the CDC Director (n = 382, 53%) or the NIH Director (n = 117, 16%) to lead the COVID-19 response (Fig 1). However, only a small percentage of participants wanted the president (n = 91, 13%) or Congress (n = 5, 1%) to lead the response (Fig 1).

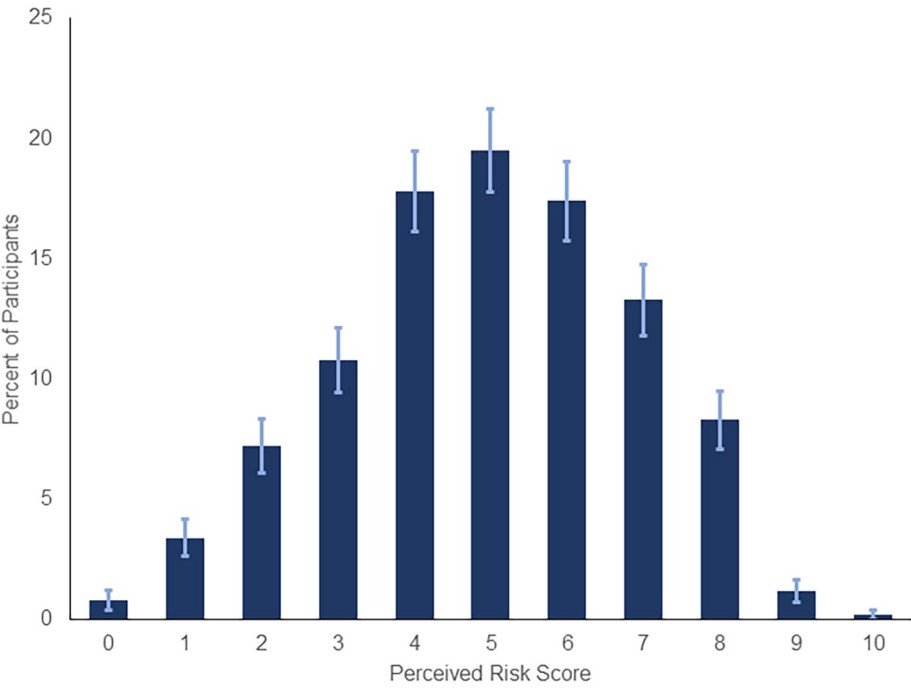

**Fig 2. Distribution of risk perception score.**

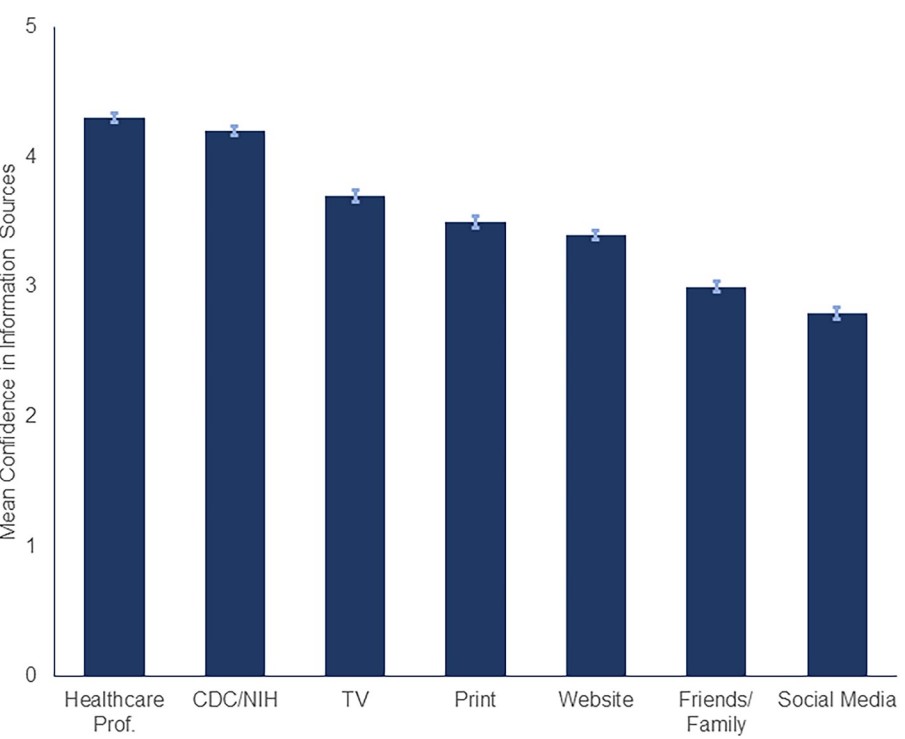

**Fig 3. Participants confidence in various information sources.**

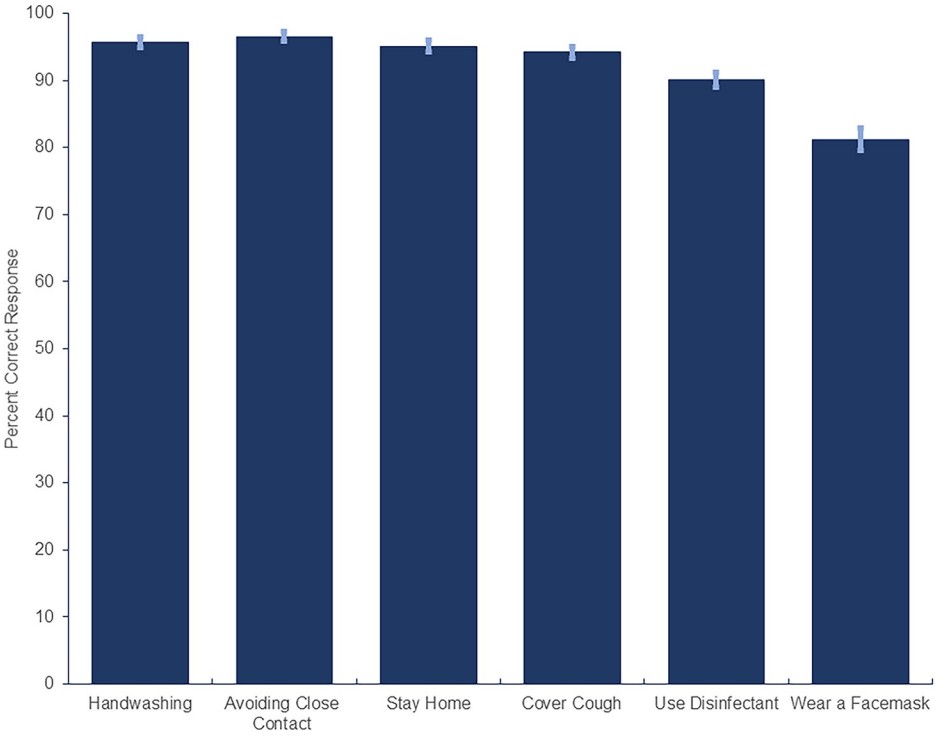

**Fig 4. Participants correctly identifying effective infection prevention measures for themselves/others.**

**Table 2. Comparison of sample result to weighted result based on age and gender.**

|                                                  | Sample Result | Weighted Result |
| ------------------------------------------------ | ------------- | --------------- |
| Risk Perception Score (mean)                     | 5.0           | 5.0             |
| Confidence in Sources of Information (mean)      |               |                 |
| Healthcare Professionals | 4.3           | 4.3             |
| CDC/NIH                   | 4.2           | 4.2             |
| TV                        | 3.7           | 3.6             |
| Print                     | 3.5           | 3.4             |
| Web                       | 3.4           | 3.4             |
| Friends/Family            | 3.0           | 3.0             |
| Social Media              | 2.8           | 2.8             |
| Who should lead the US response to COVID-19 (%)  |               |                 |
| Director of CDC           | 53.2          | 52.3            |
| Director of NIH           | 16.3          | 16.6            |
| President                 | 12.7          | 13.5            |
| Secretary of DHHS         | 9.3           | 9.6             |
| State Departments of Health | 3.5         | 3.4             |
| Local Health Departments  | 2.8           | 2.5             |
| Congress                  | 0.7           | 0.6             |

The mean risk perception score was 5.0 out of 10 (SD = 1.9; Fig 2). Strict policies for infection prevention including quarantine (n = 571, 83%) and travel restriction (n = 519, 75%) were endorsed by most participants. Additionally, thirty-five percent of participants supported "temporary discrimination based on someone's country of origin" in case of an outbreak (n = 244, 35%).

The most trusted sources of information for the participants were healthcare professionals (M = 4.3; SD = 0.9) and health officials (e.g. CDC and NIH; M = 4.2; SD = 1.0). The least trusted source of information was social media (M = 2.8; SD = 1.2; Fig 3).

Over 90% of the participants correctly identified CDC-recommended [6] infection prevention measures (Fig 4).

## Discussion

We found that the public trusted the CDC Director to lead the COVID-19 response with trust in the public health/scientific leadership being high. Responsive, open, and respectful communication with the US population by these agencies may improve public health compliance and safety. [3] Furthermore, although participants reported relatively low risk perception, many supported restrictive policies for infection prevention. A portion of the sample also supported temporary discrimination based on someone's country of origin. These responses are concerning, and preemptive targeted messaging by the public health agencies is required to ensure a compassionate response to this outbreak. Our findings may be influenced by possible selection bias because participants needed a CloudResearch account and access to smartphone/computer to participate. However, our sample was fairly representative of the general adult US population. A weighted analysis based on age and gender demonstrate that our results are generalizable to national population (Table 2). Data for weighted analysis were extracted from US Census data. [7]

## Conclusion

Given our results, the public health/scientific leadership should be at the forefront of the COVID-19 response to promote trust. Strategic messaging by the CDC and the NIH through

television, print, and internet has strong potential to alleviate unnecessary fear among the US population.

## Supporting information

**S1 Survey. Perceptions regarding the novel coronavirus outbreak questionnaire.**
(DOCX)

**S1 Data.**
(XLSX)

**S2 Data.**
(XLS)

## Author Contributions

**Conceptualization:** SarahAnn M. McFadden, Amyn A. Malik, Saad B. Omer.

**Data curation:** SarahAnn M. McFadden, Amyn A. Malik.

**Formal analysis:** SarahAnn M. McFadden, Amyn A. Malik, Obianuju G. Aguolu, Kathryn S. Willebrand.

**Methodology:** Saad B. Omer.

**Supervision:** Saad B. Omer.

**Writing – original draft:** SarahAnn M. McFadden, Amyn A. Malik, Obianuju G. Aguolu, Kathryn S. Willebrand.

**Writing – review & editing:** SarahAnn M. McFadden, Amyn A. Malik, Obianuju G. Aguolu, Saad B. Omer.

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
