## [Decision Letter · Decision Letter 0]

19 Mar 2020

PONE-D-20-05428

Perceptions of the Adult US Population regarding the Novel Coronavirus Outbreak

PLOS ONE

Dear Dr. Omer,

Thank you very much for submitting your manuscript "Perceptions of the Adult US Population regarding the Novel Coronavirus Outbreak" (#PONE-D-20-05428) for review by PLOS ONE. As with all papers submitted to the journal, your manuscript was fully evaluated by academic editor (myself) and by independent peer reviewers. The reviewers appreciated the attention to an important health topic, but they raised substantial concerns about the paper that must be addressed before this manuscript can be accurately assessed for meeting the PLOS ONE criteria. Therefore, if you feel these issues can be adequately addressed, we invite you to submit a revised version of the manuscript that addresses the points raised during the review process. We can’t, of course, promise publication at that time.

We would appreciate receiving your revised manuscript by May 02 2020 11:59PM. To enhance the reproducibility of your results, we recommend that if applicable you deposit your laboratory protocols in protocols.io, where a protocol can be assigned its own identifier (DOI) such that it can be cited independently in the future. For instructions see: http://journals.plos.org/plosone/s/submission-guidelines#loc-laboratory-protocols

We look forward to receiving your revised manuscript.

Kind regards,

Abdallah M. Samy, PhD

Academic Editor

PLOS ONE

**Journal Requirements:**

3. Your ethics statement must appear in the Methods section of your manuscript. If your ethics statement is written in any section besides the Methods, please move it to the Methods section and delete it from any other section. Please also ensure that your ethics statement is included in your manuscript, as the ethics section of your online submission will not be published alongside your manuscript.

**Reviewers' comments:**

Reviewer's Responses to Questions

**Comments to the Author**

1. Is the manuscript technically sound, and do the data support the conclusions?

Reviewer #1: Yes

2. Has the statistical analysis been performed appropriately and rigorously? 

Reviewer #1: No

3. Have the authors made all data underlying the findings in their manuscript fully available?

Reviewer #1: Yes

4. Is the manuscript presented in an intelligible fashion and written in standard English?

Reviewer #1: Yes

5. Review Comments to the Author

Reviewer #1: The current manuscript McFadden, et al., (2020), demonstrated perceptions regarding Coronavirus Disease 2019 (COVID-19) outbreak in which the first appearance was in Wuhan, China causing pandemics globally recently. So, the research objectives are too important to increase the awareness and intention to easy control COVID-19 not only in US but also globally. The manuscript language is well written. The article comprises CloudResearch survey using an electronic questionnaire which in my opinion, participants easy to involve and finish the form. But, the drawbacks of this type of online consent maybe lack accuracy, ensuring comprehension and verifying identification.

Minor Comments:

1. In abstract line 1: please add Coronavirus Disease 2019 as a definition for COVID-19.

2. In abstract line 2: the risk of infection in US, I think now changed to high due to pandemic declaration (also amend in all manuscript).

3. In Background page 3, line 1: the countries infected by COVID-19 need update according to the latest WHO report (not 28 countries).

4. In Background paragraph 1: please add a sentence contain the actual numbers of infections and mortality rates globally and in US with recent reference.

5. In page 5 and 6, Figure 1, 2: there are no error bars and the statistically significant marks, also describe the statistical parameters used in analysis (type parameters in Legends).

6. PLOS authors have the option to publish the peer review history of their article (what does this mean?). If published, this will include your full peer review and any attached files.

Reviewer #1: No

---

## [Author Response · Author response to Decision Letter 0]

25 Mar 2020

March 25, 2020

Dear Dr. Samy, 

We would like to thank you and the reviewer for your careful review of this manuscript and the insightful comments to help improve our work. We have attempted to address all the concerns that have been raised. We believe that the manuscript is stronger as a result. Please see our point-by-point responses below.

On behalf of the authorship team,

Saad Omer, MBBS, MPH, PhD, FIDSA

Director | Yale Institute for Global Health

Associate Dean (Global Health Research) | Yale School of Medicine

Professor of Medicine (Infectious Diseases) | Yale School of Medicine

Adjunct Professor | Yale School of Nursing

Susan Dwight Bliss Professor of Epidemiology of Microbial Diseases | Yale School of Public Health

Responses to Reviewer’s Comments

Reviewer 1:

1. In abstract line 1: please add Coronavirus Disease 2019 as a definition for COVID-19.

Response: The sentence has been revised as requested. The sentence now reads as follows: 

“The Coronavirus Disease 2019 (COVID-19) outbreak is spreading globally.” (line 20) 

2. In abstract line 2: the risk of infection in US, I think now changed to high due to pandemic declaration (also amend in all manuscript).

Response: We amended this both in the abstract and manuscript, but we did clarify that at the time of survey administration the risk for contracting COVID-19 was thought to be low. The sentence now reads as follows: 

“Although COVID-19 has now been declared a pandemic and risk for infection in the United States (US) is currently high, at the time of survey administration the risk of infection in the US was low.” (lines 20 – 22)

3. In Background page 3, line 1: the countries infected by COVID-19 need update according to the latest WHO report (not 28 countries).

Response: We amended this by stating it has spread across the globe and updated the reference. The sentence now reads as follows: 

“The current novel coronavirus outbreak, COVID-19, has spread across the globe with hundreds of thousands infected and thousands of deaths (1).” (lines 39 – 40)

4. In Background paragraph 1: please add a sentence contain the actual numbers of infections and mortality rates globally and in US with recent reference.

Response: We added this sentence for the US with the most recent reference from the CDC. It now reads as follows: 

“At the time of this writing (March 25, 2020), there are over 54,000 cases of COVID-19 in the US with 737 deaths (2).With COVID-19 now declared a pandemic by the World Health Organization (3), it is important to understand risk perceptions about COVID-19 and trust in political and public health/scientific leadership among the US population to better inform messaging and policies (4).” (Lines 40 – 44)

5. In page 5 and 6, Figure 1, 2: there are no error bars and the statistically significant marks, also describe the statistical parameters used in analysis (type parameters in Legends).

Response: We have added the error bars to figures 1,2 and 4 and revised figure 3 so that all error bars now show SE and are consistent. As the objective of this study was to provide descriptive data on the perceptions of the US population, we did not carry out any inferential statistical testing and hence do not have a statistically significant marks on the figures to report.

---

## [Editor Report · Decision Letter 1]

2 Apr 2020

Perceptions of the Adult US Population regarding the Novel Coronavirus Outbreak

PONE-D-20-05428R1

Dear Dr. Omer,

We are pleased to inform you that your manuscript, "Perceptions of the Adult US Population regarding the Novel Coronavirus Outbreak" (PONE-D-20-05428R1), has been judged scientifically suitable for publication and will be formally accepted for publication once it complies with all outstanding technical requirements.

With kind regards,

Abdallah M. Samy, PhD

Academic Editor

PLOS ONE

---

## [Editor Report · Acceptance letter]

14 Apr 2020

PONE-D-20-05428R1 

Perceptions of the Adult US Population regarding the Novel Coronavirus Outbreak 

Dear Dr. Omer:

I am pleased to inform you that your manuscript has been deemed suitable for publication in PLOS ONE. Congratulations! Your manuscript is now with our production department. 

With kind regards,

on behalf of

Dr. Abdallah M. Samy 

Academic Editor

PLOS ONE